# CLUSTERING FOR DIRECTED GRAPHS USING PARAMETRIZED RANDOM WALK DIFFUSION KERNELS

## ABSTRACT

Clustering based on the random walk operator has been proven effective for undirected graphs, but its generalization to directed graphs (digraphs) is much more challenging. Although the random walk operator is well-defined for digraphs, in most cases such digraphs are not strongly connected, and hence the associated random walks are not irreducible, which is a crucial property for clustering that exists naturally in the undirected setting. To remedy this, the usual workaround is to either naively symmetrize the adjacency matrix or to replace the natural random walk operator by the Pagerank random walk operator, but this can lead to the loss of valuable information carried by the graph directionality and edge density. In this paper, we introduce a new clustering framework, the *Parametrized Random Walk Diffusion Kernel Clustering* (P-RWDKC), which is suitable for handling both directed and undirected graphs. P-RWDKC is based on the diffusion geometry (Coifman & Lafon, 2006) and the generalized spectral clustering framework (Sevi et al., 2022). Accordingly, we propose an algorithm that automatically reveals the cluster structure at a given scale, by considering the random walk dynamics associated with a parametrized graph operator, and by estimating its critical diffusion time. Experiments on $K$-NN graphs constructed from real-world datasets and real-world graphs, show that in most of the tested cases our clustering approach has superior performance compared to existing approaches.

## 1 INTRODUCTION

Clustering is a fundamental unsupervised learning task whose aim is to analyze and reveal the cluster structure of unlabeled datasets, and has widespread applications in machine learning, network analysis, biology, and other fields (Kiselev et al., 2017; McFee & Ellis, 2014). Clustering for data represented as a graph has been formulated in various ways. A well-established one consists in minimizing a functional of the graph-cut (Von Luxburg, 2007; Shi & Malik, 2000), leading to the spectral clustering (SC) framework and various algorithms. SC is simple and effective, but has important limitations (Nadler & Galun, 2006) that make it unreliable in a number of non-rare data regimes. Overcoming these limitations is where the focus of the machine learning community is (Tremblay et al., 2016; Zhang & Rohe, 2018; Dall'Amico et al., 2021; Sevi et al., 2022).

A well-studied clustering approach for high-dimensional data, which is of particular interest for this work, is using the operator associated with a random walk (or, in other terms, with a Markov chain), called *random walk operator*. Meilă & Shi (2001) viewed the pairwise similarities between datapoints as edge flows of a Markov chain and proposed an **ergodic** random walk interpretation of the spectral clustering. However, and beyond SC, the first to conceive the idea of turning the distance matrix between high-dimensional data into a Markov process were Tishby & Slonim (2000). More specifically, they proposed to examine the decay of mutual information during the relaxation of the Markov process. During the relaxation procedure, the clusters emerge as quasi-stable structures, and then they get extracted using the information bottleneck method (Tishby et al., 2000). Azran & Ghahramani (2006) proposed to estimate the number of data clusters by estimating the diffusion time of the random walk operator that reveals the most significant cluster structure. Lin & Cohen (2010) proposed then to find a low-dimensional embedding of the data using the truncated power iteration of a random walk operator derived from the pairwise similarity matrix.

Clustering high-dimensional data using a random walk operator through the lens of the diffusion geometry (Coifman & Lafon, 2006; Coifman et al., 2005) has also been investigated. Nadler et al. (2006) proposed a unifying probabilistic diffusion framework based on the probabilistic interpretation of SC and dimensionality reduction algorithms using the eigenvectors of the normalized graph Laplacian. Given the pairwise similarity matrix built from high-dimensional data points, they defined a distance function between any two points based on the random walk on the graph called the diffusion distance (Coifman & Lafon, 2006; Pons & Latapy, 2005) and showed that the low-dimensional representation of the data by the first few eigenvectors of the corresponding random walk operator is optimal under a certain criterion. Recently, an unsupervised clustering methodology has been proposed based on the diffusion geometry framework (Maggioni & Murphy, 2019; Murphy & Polk, 2022), where it has been demonstrated how the diffusion time of the random walk operator can be exploited to successfully cluster datasets for which $k$-means, spectral clustering, or density-based clustering methods fail.

Although there has been a constant development of clustering approaches which are based on the random walk operator, all such efforts (like those mentioned above) have been proposed for undirected graphs. Moreover, the motivation of most of them is to run forward the random walk to avoid the costly eigendecomposition of the graph Laplacian. The extension of clustering approaches based on the random walk operator to digraphs is much more subtle and challenging. As presented earlier, random walk-based clustering approaches rely either on the eigenvectors of the random walk operator or on its iterated powers. In the directed setting, the random walk is well-defined, but it is not reversible in general like in the undirected case (Levin & Peres, 2017). Consequently, the associated eigenvectors are possibly complex, which makes their interpretation and use difficult in the context of clustering. A possible workaround would be to consider the approach based on the iterated powers of the random walk operator which would allow one to obtain a real-valued embedding and thus avoid the use of complex eigenvectors.

However, a second much more subtle and problematic bottleneck arises: the random walk's irreducibility. While in the undirected case, the random walk is irreducible, i.e. any graph vertex can be reached from any other vertex, this is not the case in general for random walks on digraphs. To overcome the irreducibility issue, either a symmetrization procedure is usually employed in the case where the digraph is derived by a $K$-NN graph construction or the original random walk operator is replaced by the operator of the *Pagerank* or *teleporting random walk* (Page et al., 1999). For these two workarounds, valuable information can potentially be discarded. To address this problem, we present a new clustering algorithm based on the diffusion geometry framework, which is suitable for any digraph, either obtained by $K$-NN constructions or by real digraphs representing asymmetric relationships between vertices (e.g. citation graphs). Our approach stems from a new type of graph Laplacians (Sevi et al., 2022) that is parametrized by an arbitrary vertex measure, capable of encoding digraph information, and from which we derive a random walk operator. Besides, we can exploit the necessary diffusion time of this parametrized random walk to successfully reveal clusters at different scales.

This work is part of the main line of work on *edge density-based clustering* on digraphs that seeks clusters characterized by high intra-cluster and low inter-cluster edge densities, and has produced several methods in the last two decades Zhou et al. (2005); Meilă & Pentney (2007); Satuluri & Parthasarathy (2011); Rohe et al. (2016); Palmer & Zheng (2020); Sevi et al. (2022); Klus & Djurdjevac Conrad (2023). Lately, there have been proposed approaches Cucuringu et al. (2020); Laenen & Sun (2020); Coste & Stephan (2021); Hayashi et al. (2022), which are called *flow-based* and whose objective is opposite to the traditional density-based graph clustering view-point.

The contribution of this work is manifold: i) We propose a new similarity kernel operator, which we term ***random walk diffusion kernel*** (RWDK), that derives directly from the original definition of the diffusion distance. This latter allows us to theoretically justify the use of the random walk operator as a fundamental graph operator for clustering. ii) We generalize this kernel by proposing the use of a parametrized random walk operator, which allows us to extend the diffusion distance to digraphs. iii) From there, we present our clustering algorithm on digraphs, the ***parametrized random walk diffusion kernel clustering***, based on the parametrized RWDK operator considered as a data embedding. iv) We propose a general method for estimating the diffusion time that best reveals a given number of clusters. Finally, we show that our approach is efficient on both synthetic and real-world datasets, as it outperforms existing methods in most of the tested cases.

## 2 BACKGROUND

**Graph theory and graph functions.** Let $\mathcal{G} = (\mathcal{V}, \mathcal{E}, w)$ be a weighted directed graph (digraph), where $\mathcal{V}$ is the finite set of $N = |\mathcal{V}|$ vertices, and $\mathcal{E} \subseteq \mathcal{V} \times \mathcal{V}$ is the finite set of edges. Each edge $(i, j)$ is an ordered vertex pair indicating the direction of a link from vertex $i$ to vertex $j$. With little abuse of notation we sometimes use $i \in \mathcal{V}$ implying a vertex index $i \in [1, ..., N]$. The edge weight function $w : \mathcal{V} \times \mathcal{V} \to \mathbb{R}_+$ associates a nonnegative real value to every vertex pair: $w(i, j) > 0$, iff $(i, j) \in \mathcal{E}$, otherwise $w(i, j) = 0$. A digraph $\mathcal{G}$ is represented by a weighted adjacency matrix $\mathbf{W} = \{w(i, j)\}_{i,j=1}^N \in \mathbb{R}_+^{N \times N}$, where $w(i, j)$ is the weight of the edge $(i, j)$. We define the out-degree and the in-degree of the $i$-th vertex by $d_{\mathrm{out},i} = \sum_{j=1}^N w(i, j)$ and $d_{\mathrm{in},i} = \sum_{j=1}^N w(j, i)$, respectively. Also, the function $\mathbf{D}_\nu = \mathrm{diag}(\nu)$ is a square diagonal matrix with the elements of the input vector $\nu$ in its diagonal. Consider a graph function $f$ mapping all of its vertices to an $N$-dimensional vector: $f = [\, f(i) \,]_{\forall i \in \mathcal{V}}^{\mathsf{T}} \in \mathbb{R}^N$. We assume that graph functions are defined in $\ell^2(\mathcal{V}, \nu)$, which is the Hilbert space of functions, defined over the vertex set $\mathcal{V}$ of $\mathcal{G}$, endowed with the inner product associated with an arbitrary positive measure $\nu$. Let $\delta_i \in \{0, 1\}^{N \times 1}$ be the vector output of the Kronecker delta function at $i \in \mathcal{V}$. Any function $\nu : \mathcal{V} \to \mathbb{R}_+$, associating a nonnegative value to each graph vertex, can be regarded as a positive vertex measure.

**Random walk fundamentals.** What we call in short a random walk on a weighted graph $\mathcal{G}$, is defined more formally as a natural random walk on the graph, which is a homogeneous Markov chain $\mathcal{X} = (X_t)_{t \geq 0}$ with a finite state space $\mathcal{V}$, and with state transition probabilities proportional to the edge weights. The entries of the transition matrix $\mathbf{P} = [\, p(i, j) \,]_{\forall i,j \in \mathcal{V}}$ are defined by:

$$p(i, j) = \mathbb{P}(X_{t+1} = j \mid X_t = i) = \frac{w(i, j)}{\sum_{z \in \mathcal{V}} w(i, z)}.$$

Algebraically, the transition matrix $\mathbf{P} \in \mathbb{R}^{N \times N}$ can be expressed as $\mathbf{P} = \mathbf{D}_{\mathrm{out}}^{-1} \mathbf{W}$, $\mathbf{D}_{\mathrm{out}} = \mathbf{D}_{d_{\mathrm{out}}}$ (respectively, $\mathbf{D}_{\mathrm{in}} = \mathbf{D}_{d_{\mathrm{in}}}$)whose spectrum $\mathrm{sp}(\mathbf{P}) \in [-1, 1]$. For a strongly connected digraph $\mathcal{G}$, the random walk $\mathcal{X}$ is irreducible. If, in addition, the irreducible random walk $\mathcal{X}$ admits the aperiodicity condition, then $\mathcal{X}$ is ergodic, and therefore as $t \to \infty$, the probability measures $p_t(i, *) = \delta_i^{\mathsf{T}} \mathbf{P}^t$, for all $i \in \mathcal{V}$, converge toward a *unique* stationary distribution denoted by the row vector $\pi \in \mathbb{R}_+^N$ (Brémaud, 2013). A random walk $\mathcal{X}$ with stationary distribution $\pi$ and transition matrix $\mathbf{P}$ is called reversible if $\pi(i)p(i, j) = \pi(j)p(j, i)$, for all $i, j \in \mathcal{V}$. Within the undirected setting: $d_{\mathrm{out},i} = d_{\mathrm{in},i} = d_i$, where $d \in \mathbb{R}_+^{N \times 1}$ is the vector of the vertex degrees, also and $\mathbf{D}_d = \mathrm{diag}(d)$ is a square diagonal matrix with the elements of the input vector $d$ in its diagonal. Moreover, the stationary distribution is proportional to the vertex degree distribution, i.e. $\pi \propto d$.

**Parametrized graph Laplacians.** A notable example of parametrized operators that we use later, is the generalized graph Laplacian from Sevi et al. (2022), which is a new type of graph operators for (but not restricted to) digraphs. Relevant theoretical background is provided in Appendix A.4.1.

**Definition 2.1.** *Generalized graph Laplacians (Sevi et al., 2022). Let $\mathbf{P}$ be the transition matrix of a random walk on a digraph $\mathcal{G}$. Under an arbitrary positive vertex measure $\nu$ on $\mathcal{G}$, consider the positive vertex measure $\xi = \nu^{\mathsf{T}} \mathbf{P}$. Let also the diagonal matrices $\mathbf{D}_\nu = \mathrm{diag}(\nu)$, $\mathbf{D}_\xi = \mathrm{diag}(\xi)$ and $\mathbf{D}_{\nu+\xi} = \mathrm{diag}(\nu + \xi)$. Then, we can define three generalized Laplacians of $\mathcal{G}$ as follows:*

$$\textit{generalized random walk Laplacian:} \quad \mathbf{L}_{\mathrm{RW},(\nu)} = (\mathbf{D}_{\nu+\xi})^{-1}(\mathbf{D}_\nu \mathbf{P} + \mathbf{P}^{\mathsf{T}} \mathbf{D}_\nu) \tag{1}$$

$$\textit{unnormalized generalized Laplacian:} \quad \mathbf{L}_{(\nu)} = \mathbf{D}_{\nu+\xi} - (\mathbf{D}_\nu \mathbf{P} + \mathbf{P}^{\mathsf{T}} \mathbf{D}_\nu) \tag{2}$$

$$\textit{normalized generalized Laplacian:} \quad \mathcal{L}_{(\nu)} = \mathbf{D}_{\nu+\xi}^{-1/2} \mathbf{L}_{(\nu)} \mathbf{D}_{\nu+\xi}^{-1/2}. \tag{3}$$

$\mathbf{L}_{\mathrm{RW},(\nu)}$, $\mathbf{L}_{(\nu)}$, and $\mathcal{L}_{(\nu)}$ are parametrized by an arbitrary vertex measure has the modeling capacity to encode the graph directionality and edge density using the random walk dynamics of the original digraph. When the transition matrix $\mathbf{P}$ is irreducible and the vertex measure $\nu$ is the ergodic measure $\pi$, they correspond to the directed graph Laplacians in Chung (2005).

## 3 PARAMETRIZED RANDOM WALK OPERATOR ON GRAPHS

In this section, we introduce the parametrized random walk operator that we propose, and we show how it is derived from the generalized graph Laplacian (presented at the end of Sec. 2).

**Definition 3.1.** *Parametrized random walk operator* (P-RW). *Let* $\mathbf{P}$ *be the transition matrix of a random walk on a digraph* $\mathcal{G}$. *Under an arbitrary positive vertex measure* $\nu$ *on* $\mathcal{G}$, *consider the positive vertex measure* $\xi = \nu^\mathsf{T}\mathbf{P}$. *Let also the diagonal matrices* $\mathbf{D}_\nu = \mathrm{diag}(\nu)$, $\mathbf{D}_\xi = \mathrm{diag}(\xi)$ *and* $\mathbf{D}_{\nu+\xi} = \mathrm{diag}(\nu+\xi)$. *Finally, let* $\mathcal{X}_\nu$ *be the random walk on* $\mathcal{G}$ *with the associated random walk operator (transition matrix)* $\mathbf{P}_{(\nu)}$ *defined by:*

$$\mathbf{P}_{(\nu)} = (\mathbf{D}_{\nu+\xi})^{-1}(\mathbf{D}_\nu\mathbf{P} + \mathbf{P}^\mathsf{T}\mathbf{D}_\nu). \tag{4}$$

It is easy to verify that $\mathbf{P}_{(\nu)}$ is a transition matrix (see Appendix A.1.2) and that there is the following relation with generalized random walk Laplacian.

**Definition 3.2.** *Relation between* $\mathbf{L}_{\mathrm{RW},(\nu)}$ *and* $\mathbf{P}_{(\nu)}$. *Let* $\mathbf{P}$ *be the transition matrix of a random walk on a digraph* $\mathcal{G}$. *Under an arbitrary positive vertex measure* $\nu$ *on* $\mathcal{G}$. *The generalized random walk Laplacian and the parametrized random walk on* $\mathcal{G}$ *are related as follows:*

$$\mathbf{L}_{\mathrm{RW},(\nu)} = \mathbf{I} - \mathbf{P}_{(\nu)}.$$

As stated in Sevi et al. (2022), the generalized random walk Laplacian, $\mathbf{L}_{\mathrm{RW},(\nu)}$, is self-adjoint in $\ell^2(\mathcal{V}, \nu+\xi)$. Consequently, the generalized random walk $\mathbf{P}_{(\nu)}$ is also self-adjoint in $\ell^2(\mathcal{V}, \nu+\xi)$, and the associated random walk $\mathcal{X}_\nu$ is ergodic (under the aperiodicity condition) and admits the ergodic distribution $\pi_\nu$. Therefore, $\mathbf{P}_{(\nu)}$ is a random walk operator parameterized by a vertex measure $\nu$ that encodes the random walk dynamic of the original digraph.

## 4 DIFFUSION GEOMETRY FOR DIGRAPHS

In this section, we review the concept of diffusion geometry (Coifman & Lafon, 2006), and we show: i) how its core feature, the *diffusion distance*, can be expressed as a Mahalanobis distance involving a specific kernel matrix, which we call *Random Walk Diffusion Kernel* (RWDK); ii) how the SC's diagonalization step can be thought of as a function applied to the spectrum of the Laplacian and the conceptual connection with the RWDK, and iii) accordingly a clustering algorithm.

### 4.1 THE DIFFUSION DISTANCE AS A MAHALANOBIS DISTANCE

The seminal work by (Coifman & Lafon, 2006) introduced the *diffusion geometry framework*, which uses diffusion processes as basic tool to find meaningful geometric descriptions for dataset. The framework can provide different geometric representations of the dataset by iterating the Markov transition matrix, which is equivalent to running forward the random walk. The key element of the diffusion geometry is the diffusion distance (Coifman & Lafon, 2006; Pons & Latapy, 2005) defined as follows. We provide relevant theoretical background in Appendix A.4.2.

**Definition 4.1.** *Diffusion distance.* *Let* $\mathbf{P}$ *be the transition matrix of a reversible random walk on an undirected graph* $\mathcal{G}$, *with an ergodic distribution* $\pi$. *Let also* $\|f\|_{1/\pi}^2 = \langle f, \mathbf{D}_\pi^{-1} f \rangle$, *with* $\mathbf{D}_\pi = \mathrm{diag}(\pi)$, *be the* $\ell^2$*-norm of a graph function* $f$ *induced by the measure* $1/\pi$. *The diffusion distance at time* $t \in \mathbb{N}$ *between the vertices* $i$ *and* $j$ *is defined by:*

$$d_t^2(i,j) = \|p_t(i,*) - p_t(j,*)\|_{1/\pi}^2. \tag{5}$$

As noted in (Coifman & Lafon, 2006), the diffusion distance emphasizes the cluster structure, if present. Next, we show that the diffusion distance can be seen as a Mahalanobis distance.

**Proposition 4.1.** *Diffusion distance as a Mahalanobis distance.* *Let* $\mathbf{P}$ *be the transition matrix of a reversible random walk on an undirected graph* $\mathcal{G}$, *with an ergodic distribution* $\pi$. *The diffusion distance* $d_t^2(i,j)$ *between vertices* $i$ *and* $j$, *at a given time* $t \in \mathbb{N}$, *can be expressed as the following Mahalanobis distance:*

$$d_t^2(i,j) = (\delta_i - \delta_j)^\mathsf{T}\mathbf{K}_t(\delta_i - \delta_j),$$

*where the similarity positive definite kernel matrix* $\mathbf{K}_t$ *is defined by:*

$$\mathbf{K}_t = \mathbf{P}^{2t}\mathbf{D}_d^{-1}. \tag{6}$$

The diffusion distance reveals a similarity kernel matrix $\mathbf{K}_t$ that we call ***Random Walk Diffusion Kernel*** (**RWDK**), which is simply a power of the transition matrix $\mathbf{P}$ normalized by the vertex degrees. Consequently, using the diffusion distance $d_t^2$ is equivalent to using the RWDK matrix $\mathbf{K}_t$.

### 4.2 RANDOM WALK DIFFUSION KERNEL, NORMALIZED GRAPH LAPLACIAN AND SPECTRAL CLUSTERING

Spectral clustering (SC) is one of the most widely used clustering methods due to its simplicity, efficiency, and strong theoretical foundation (Shi & Malik, 2000; Ng et al., 2002; Von Luxburg, 2007; Peng et al., 2015; Boedihardjo et al., 2021). Given a fixed number of clusters $k$, SC consists of three main steps: i) construct the graph Laplacian matrix; ii) compute the eigenvectors associated with the $k$ smallest eigenvectors of the Laplacian matrix and store them as columns in a matrix; iii) apply $k$-means on the rows of the latter matrix, which are regarded as embedded representations of the datapoints. *We aim to highlight how the SC's second step and the RWDK do have similar characteristics.* Let us consider the normalized graph Laplacian matrix $\mathcal{L}$ (Chung & Graham, 1997) with eigendecomposition $\mathcal{L} = \sum_{j=1}^{N} \vartheta_j \phi_j \phi_j^\mathsf{T}$, ordered eigenvalues $0 \leq \vartheta_1 \leq ... \leq \vartheta_N \leq 2$ and eigenvectors $\{\phi_j\}_{j=1}^{N}$. Computing the eigenvectors associated to the $k$ smallest eigenvalues of $\mathcal{L}$ amounts to applying a function $f_1$ on the spectrum matrix of $\mathcal{L}$ such that $f_1(x) = 1$ if $x \leq \vartheta_k$, and $f_1(x) = 0$ otherwise, namely $\mathbf{H} = f_1(\mathcal{L}) = \sum_{j=1}^{k} \vartheta_j \phi_j \phi_j^\mathsf{T}$. Thanks to the relation $\mathcal{L} = \mathbf{D}^{\frac{1}{2}}(\mathbf{I} - \mathbf{P})\mathbf{D}^{-\frac{1}{2}}$, selecting the $k$ smallest eigenvalues of $\mathcal{L}$ is thus equivalent to selecting the $k$ largest eigenvalues of $\mathbf{P}$, namely $\mathbf{H} = f_1(\mathcal{L}) = \mathbf{D}^{\frac{1}{2}} f_1(\mathbf{I} - \mathbf{P})\mathbf{D}^{-\frac{1}{2}}$. On the other hand, the RWDK $\mathbf{K}_t$ corresponds to applying a function $f_2(x, t) = x^{2t}$ on the spectrum matrix of $\mathbf{P}$. As $t \to \infty$, $f_2(\mathbf{P}, t) \to \pi \mathbf{1}^\mathsf{T}$, and consequently $\mathbf{K}_t = f_2(\mathbf{P}, t)\mathbf{D}^{-1} \to [\text{tr}(\mathbf{D}_d)]^{-1}\mathbf{1}\mathbf{1}^\mathsf{T}$. As $t$ increases, $f_2(\mathbf{P})$ becomes smoother because $f_2$ acts as a soft truncation of the spectrum matrix of $\mathbf{P}$, which preserves those eigenvectors of $\mathbf{P}$ that are associated with the largest eigenvalues. However, as $t$ increases, $f_1$ acts as a hard truncation of the spectrum matrix of $\mathcal{L}$, which preserves the $k$ eigenvectors associated with the lowest ones up to $\vartheta_k$. Consequently, the RWDK $\mathbf{K}_t$ is an adaptive alternative to $\mathbf{H}$, and the aim is to determine the iteration time $t$ that best reveals the $k$ clusters the user looks for.

### 4.3 PARAMETRIZED RANDOM WALK KERNEL CLUSTERING

We have shown that the diffusion distance $d_t^2$ is directly related to the RWDK and that for a given diffusion time $t$, the RWDK is an alternative to computing the eigenvectors of the graph Laplacian. However, the original diffusion distance defined in Eq. 5 was settled in the undirected setting w.r.t a transition matrix associated with a reversible random walk. For an arbitrary transition matrix $\mathbf{P}$ and an arbitrary measure $\mu$, the diffusion distance between vertices $i$ and $j$, at a given diffusion time $t$, is indeed defined as the weighted Euclidean distance between the rows of $\mathbf{P}^t$: $d_t^2(i, j, \mu) = \|p_t(i, *) - p_t(j, *)\|_\mu^2 = (\delta_i - \delta_j)^\mathsf{T}\mathbf{K}_{(t,\mu)}(\delta_i - \delta_j)$. We have established the last expression in Sec.4.1; it suggests that one can see the diffusion distance as a Mahalanobis distance involving a similarity kernel $\mathbf{K}_{(t,\mu)} = \mathbf{P}^t\mathbf{D}_\mu(\mathbf{P}^t)^\mathsf{T}$ with $\mathbf{D}_\mu = \text{diag}(\mu)$.

By breaking down this way $d_t^2\mathbf{K} = \mathbf{P}^{2t}\mathbf{D}^{-1}$ (see Eq. 6), $\mathbf{P}$ needs to be the transition matrix of a reversible random walk with stationary measure $\pi$, and the vertex measure needs to be $\mu = 1/\pi$. This is the reason why the original diffusion distance formulation was only used in the undirected setting. Now, the advantage of the parametrized random walk operator we use, is that it is also a reversible transition matrix for digraphs, and hence enables naturally the extension of the diffusion distance for digraphs.

**Definition 4.2.** *Parametrized diffusion distance. Let $\mathcal{X}$ be a random walk on digraph $\mathcal{G}$, with transition matrix $\mathbf{P}$. Let $\nu$ be an arbitrary positive vertex measure on $\mathcal{G}$, and $\xi$ be the vertex measure defined by $\xi = \nu^\mathsf{T}\mathbf{P}$. Define the diagonal matrices $\mathbf{D}_\nu = \text{diag}(\nu)$ and $\mathbf{D}_\xi = \text{diag}(\xi)$. On $\mathcal{G}$, let $\mathcal{X}_\nu$ be a random walk associated with the random walk operator $\mathbf{P}_{(\nu)}$ parametrized by an arbitrary measure $\nu$ defined in Eq. 4 and ergodic measure $\pi_\nu$. Let $p_{t,\nu}(i, *) = \delta_i^\mathsf{T}\mathbf{P}_{(\nu)}^t$ be the conditional probability vector given the vertex $i$ at the diffusion time $t$. The parametrized diffusion distance between the vertices $i$ and $j$, at a given diffusion time $t$, is defined by:*

$$d_{t,\nu}^2(i, j) = \|p_{t,\nu}(i, *) - p_{t,\nu}(j, *)\|_{1/\pi_\nu}^2 = (\delta_i - \delta_j)^\mathsf{T}\mathbf{K}_{(t,\nu)}(\delta_i - \delta_j), \tag{7}$$

*with $\mathbf{K}_{(t,\nu)}$ is the **parametrized random walk diffusion kernel** (P-RWDK) defined as:*

$$\mathbf{K}_{(t,\nu)} = \mathbf{P}_{(\nu)}^t \mathbf{D}_{\nu+\xi}^{-1}. \tag{8}$$

Finally, the normalized generalized Laplacian $\mathcal{L}_{(\nu)}$ and the parametrized random walk operator $\mathbf{P}_{(\nu)}$ are related in the same manner the normalized Laplacian relates to the random walk operator (see

---

**Algorithm 1** Parametrized Random Walk Diffusion Kernel Clustering (P-RWDKC)

---

**Input:** $\mathbf{W} \in \mathbb{R}^{N \times N}$: adjacency matrix, $k$: number of clusters, $\nu$: vertex measure, $t_d$: diffusion time
**Output:** $\boldsymbol{V}_{t_d}$: graph $k$-partition for diffusion time $t_d$

---

1: Compute the parametrized random walk operator $\mathbf{P}_{(\nu)}$, see Eq. 4
2: Compute the parametrized random walk diffusion kernel $\mathbf{K}_{(t_d, \nu)}$, see Eq. 8
3: Consider each $x_i \in \mathbb{R}^N$, $i = 1, ..., N$, to be the embedding of the $i$-th vertex, represented by the $i$-th row of $\mathbf{K}_{(t_d, \nu)}$, and apply a clustering method (e.g. $k$-means) to all these vectors asking for $k$ clusters
4: Obtain the $k$-partition $\boldsymbol{V}_{t_d} = \{V_{t_d}\}_{j=1}^{k}$ of the graph vertices based on the clustering result of Step 3
5: **return** $\boldsymbol{V}_{t_d}$

---

Sec.4.2), and therefore we can write: $\mathcal{L}_{(\nu)} = \mathbf{D}_{\nu+\xi}^{1/2}(\mathbf{I} - \mathbf{P}_{(\nu)})\mathbf{D}_{\nu+\xi}^{-1/2}$. Consequently, the same approach of Sec.4.2 is also valid in this setting. Once we have defined the parameterized diffusion distance $d_{(t,\nu)}^2$, we are in position to present the novel ***parametrized random walk diffusion kernel clustering*** (**P-RWDKC**) for both undirected and directed graphs (Alg. 1), which is based on the parametrized RWDK $\mathbf{K}_{(t,\nu)}$.

## 5 THE P-RWDKC METHOD IN PRACTICE

In the previous section, we described a novel and simple clustering algorithm for digraphs based on an operator that is parametrized by an arbitrary vertex measure $\nu$, and a its diffusion up to time $t$. To render our algorithm more flexible for practical use, two important aspects points need to be addressed: i) the design of the vertex measure $\nu$, ii) the estimation of the diffusion time.

### 5.1 DESIGNING THE VERTEX MEASURE

Designing the vertex measure is one of the major aspects of P-RWDKC as we aim to capture with it the random walk dynamics of the original digraph in our parametrized random walk operator. To do so, we propose a vertex measure derived from the iterated powers of a random walk consisting of the forward and backward digraph's flow information. Specifically, the proposed vertex measure can be parametrized by three parameters ($t \in \mathbb{N}$, $\gamma \in [0, 1]$, $\alpha \in \mathbb{R}$) and is formally given by:

$$\nu_{(t,\gamma)}^{\alpha}(i) = \left(\tfrac{1}{N} \mathbf{1}_{N \times 1}^{\mathsf{T}} \mathbf{P}_{\gamma}^{t} \delta_i\right)^{\alpha}, \tag{9}$$

where $\mathbf{1}_{N \times 1}$ is the all-ones vector, recall that $\delta_i \in \{0, 1\}^{N \times 1}$ is the vector returned by the Kronecker delta function at $i \in \mathcal{V}$, and

$$\mathbf{P}_{\gamma} = \gamma \mathbf{P}_{\text{out}} + (1 - \gamma) \mathbf{P}_{\text{in}}, \tag{10}$$

where $\mathbf{P}_{\text{in}} = \mathbf{D}_{\text{in}}^{-1} \mathbf{W}^{\mathsf{T}}$ and $\mathbf{P}_{\text{out}} = \mathbf{D}_{\text{out}}^{-1} \mathbf{W}$ (recall that $\mathbf{D}_{\text{out}} = \mathbf{D}_{d_{\text{out}}}$ and $\mathbf{D}_{\text{in}} = \mathbf{D}_{d_{\text{in}}}$).

The three parameters have an easy-to-see role, and their use is optional as one can set them to values that neutralize their effect (i.e. $t = 1$, $\alpha = 1$, $\gamma = 0.5$). The random walk iteration parameter $t$ controls the diffusion time, $\gamma$ controls the mixing between $\mathbf{P}_{\text{out}}$ (forward information) and $\mathbf{P}_{\text{in}}$ (backward information), and $\alpha$ controls the re-weighting of the vertex measure. Plugging $\nu_{(t,\gamma)}^{\alpha}$ to Eq. 4 yields the following expression for the parametrized random walk $\mathbf{P}_{(\nu_{(t,\gamma)}^{\alpha})}$ and hence the P-RWDK $\mathbf{K}_{(t_d, \nu_{(t,\gamma)}^{\alpha})}$. Note that in our implementation, we use $\tilde{\mathbf{P}}_{\gamma} = (\mathbf{I} + \mathbf{P}_{\gamma})/2$ instead of $\mathbf{P}_{\gamma}$.

The choice of the vertex measure can have a significant influence on the clustering performance. Intuitively, one would be interested to see how "concentrated" are the measure's configurations that lead to good clusterings. When the problem is easy (well-separated clusters), the influence of the vertex measure is very limited and multiple different measure parametrizations may lead to good clustering (and many different algorithms may do the same). On the other hand, in difficult cases (clusters that are intricate and/or imbalanced and/or sparse) the vertex measure will be key to reaching high clustering performance, and maybe more "concentrated" in a region of good parametrizations in $\mathbb{R}^N$. We can also see the impact of the measure by considering that vanilla SC can perform arbitrarily badly. This has been partially studied theoretically in Sevi et al. (2022) and is also shown empirically in our experimental results in Sec. 6.

## 5.2 Determining the appropriate diffusion time in the unsupervised setting

For a known number of clusters $k$, we aim at determining the best diffusion time $t_d$ for the P-RWDKC algorithm. As we work in an unsupervised setting, and hence lacking ground truth, determining the right diffusion time is challenging (Shan & Daubechies, 2022). Our main insights to deal with this matter are derived from the concepts of diffusion geometry (Coifman & Lafon, 2006), the theory of nearly uncoupled Markov chains (Tifenbach, 2011; Sharpe & Wales, 2021) and the metastability of Markov chains (Landim & Xu, 2015).

As stated in (Coifman & Lafon, 2006), assuming that the graph has clusters and/or a multi-scale structure, the random walk diffusion reveals clusters at key moments of the diffusion at the course of time. The emergence of clusters can be understood through the prism of metastability theory: for a given cluster structure, there is a critical time-scale when an irreducible random walk becomes nearly reducible, diffusing only inside clusters. Typically, this means to observe approximately that, at that diffusion time, our operator enjoys high intra-cluster compactness and high inter-cluster separation. Cluster validity indexes can be employed for measuring the ratio of the former to the latter of the quantities (José-García & Gómez-Flores, 2021). Here, we propose the use of the Calinski–Harabasz criterion (CH; also known as Variance ratio criterion) (Caliński & Harabasz, 1974), which computes the ratio between the distance of the cluster centroids to the global centroid, and the distance of the datapoints of each cluster to its cluster centroid.

To compute this variance criterion, we propose two types of distances between data points, either the Euclidean distance between original data points, or a distance between their embedded representations, which in the context of this work means to consider the rows of a reference graph operator. Given the set of $N$ datapoints $\boldsymbol{X} = \{\boldsymbol{x}_1, ..., \boldsymbol{x}_N\}$ partitioned into $k$ clusters, denoted by $\boldsymbol{V} = \{V_j\}_{j=1}^k$, we denote by $\boldsymbol{\mu}_j = \frac{1}{|V_j|} \sum_{\boldsymbol{x}_i \in V_j} \boldsymbol{x}_i$ the centroid of cluster $j$, by $\boldsymbol{\mu} = \frac{1}{N} \sum_{x_i \in \boldsymbol{X}} \boldsymbol{x}_i$ the centroid of $\boldsymbol{X}$, and by $d(\boldsymbol{x}_i, \boldsymbol{x}_j)$ the pairwise distance used for two datapoints $\boldsymbol{x}_i, \boldsymbol{x}_j \in \boldsymbol{X}$. Thus, the CH criterion endowed with a given distance $d$ is defined by:

$$\text{CH}(\boldsymbol{X}, \boldsymbol{V}) = \frac{N - k}{k - 1} \frac{\sum_{j=1}^k |V_j| \, d(\boldsymbol{\mu}_j, \boldsymbol{\mu})}{\sum_{j=1}^k \sum_{\boldsymbol{x}_i \in V_j} d(\boldsymbol{x}_i, \boldsymbol{\mu}_j)}.$$

In the standard case, where we consider multidimensional data vectors, $\boldsymbol{X} = \{\boldsymbol{x}_1, ..., \boldsymbol{x}_N\} \in \mathbb{R}^d$, the usual CH criterion endowed with the Euclidean distance $d(\boldsymbol{x}_i, \boldsymbol{x}_j) = \|\boldsymbol{x}_i - \boldsymbol{x}_j\|^2$ is usually used. However, there are many reasons that the usual CH criterion may be either not directly applicable (e.g. when the input is only a graph and no point cloud), or inefficient (e.g. when the clusters of the input point cloud are non-convex or nested, and the Euclidean distance cannot help distinguishing them). For this reason, as part of our framework, we propose to extend the CH criterion for this setting, based on the Kullback–Leibler divergence as a distance Van Erven & Harremos (2014).

**Definition 5.1.** *Probability Density-based Calinski-Harabasz (DCH) criterion. Let $\mathcal{G} = (\mathcal{V}, \mathcal{E})$ be a digraph with cardinality $|\mathcal{V}| = N$. Let $\mathbf{P}$ be the transition matrix of a random walk on $\mathcal{G}$. The digraph $\mathcal{G}$ is partitioned into $k$ clusters denoted by $\boldsymbol{V} = \{V_j\}_{j=1}^k$. Let $\boldsymbol{p}(i, *) = \delta_i^\mathsf{T} \mathbf{P}$ be the conditional probability vector given the vertex $i$ that we consider as the representation of the vertex $i$. We define as reference data representation $\boldsymbol{X} = \{\boldsymbol{p}(i, *) \in \mathbb{R}^N\}_{i=1}^N$ the set of conditional probability vectors associated with the vertices of $\mathcal{G}$. Let us denote by $\boldsymbol{\mu}_j = \frac{1}{|V_j|} \sum_{\boldsymbol{p}(i,*) \in V_j} \boldsymbol{p}(i, *)$ the centroid of cluster $j$, and $\boldsymbol{\mu} = \frac{1}{N} \sum_{\boldsymbol{p}(i,*) \in \boldsymbol{X}} \boldsymbol{p}(i, *)$ the centroid of $\boldsymbol{X}$. The Kullback–Leibler divergence between two discrete probability distributions, $\boldsymbol{p}$ and $\boldsymbol{q}$, is defined as $\mathcal{D}_{\text{KL}}(\boldsymbol{p}, \boldsymbol{q}) = \sum_y p(y) \log \frac{p(y)}{q(y)}, \text{s.t. } q(y) \neq 0$. Given a set of datapoints $\boldsymbol{X}$ and a partition $\boldsymbol{V}$, the probability density-based Calinski-Harabasz (DCH) criterion endowed with the Kullback–Leibler divergence $\mathcal{D}_{\text{KL}}(\boldsymbol{p}, \boldsymbol{q})$, is defined by:*

$$\text{DCH}(\boldsymbol{X}, \boldsymbol{V}) = \frac{N - k}{k - 1} \frac{\sum_{j=1}^k |V_j| \, \mathcal{D}_{\text{KL}}(\boldsymbol{\mu}_j, \boldsymbol{\mu})}{\sum_{j=1}^k \sum_{\boldsymbol{p}(i,*) \in V_j} \mathcal{D}_{\text{KL}}(\boldsymbol{p}(i, *), \boldsymbol{\mu}_j)}.$$

We thus estimate the diffusion time in practice by evaluating the CH or DCH criterion for the partitions associated with the dyadic powers of the parametric random walk $\mathbf{P}_{(\nu)}^{2^j}$ with $\nu = \mathbf{1}$ (the uniform measure) and $j \in \{0, ..., J\}$ with $J = 15$. Since the purpose is to estimate the diffusion time, taking the vertex measure equal to a uniform measure is sufficient for this situation. We summarize this procedure in Alg. 2.

---

**Algorithm 2** Estimating the random walk diffusion time $t^*$

---

**Input:** Reference representation $\boldsymbol{X}$; $\mathbf{W}$: adjacency matrix; $k$: number of clusters; $J$: max number of iterations
**Output:** $t^*$: the estimated diffusion time that best reveals $k$ clusters

---

1: Set $\nu = \mathbf{1}$ for the vertex measure
2: Compute the parametrized random walk operator $\mathbf{P}_{(\nu)}$, see Eq. 4
3: **for** $j = 0$ **to** $J$ **do**
4:      Apply $k$-means on $\mathbf{P}_{(\nu)}^{2^j}$
5:      Obtain the $k$-partition $\boldsymbol{V}_j = \{V_{q,j}\}_{q=1}^k$ of the graph vertices based on the clustering result of Step 4.
6: **end for**
7: Select $j^\star = \underset{j \in \{0,...,J\}}{\operatorname{argmax}} \; \mathrm{CH}\big(\boldsymbol{X}, \boldsymbol{V}_j\big) \; \big(\text{or } \mathrm{DCH}\big(\boldsymbol{X}, \boldsymbol{V}_j\big)\big)$.
8: **return** $t^* = 2^{j^\star}$

---

## 6    Experiments

**Setup and competitors.** In this section, we demonstrate the effectiveness of our approach both on digraphs obtained from high-dimensional data through graph construction procedures, and real-world graphs. When dealing with high-dimensional data, we use the $K$-nearest neighbor ($K$-NN) graph construction with $K = \lfloor \log(N) \rfloor$, which produces (relatively) sparse and non-strongly connected digraphs. A resulting $K$-NN graph is unweighted, directed, and represented by its non-symmetric adjacency matrix $\mathbf{W} = \{w_{ij}\}_{i,j=1}^N$, with entries $w_{ij} = \mathbb{1}\left\{\|x_i - x_j\|^2 \leq \mathrm{dist}_K(x_i)\right\}$. In the latter, $x_i, x_j \in \mathbb{R}^d$ stand for the original coordinates of the datapoints corresponding to the vertices $i$ and $j$, $\mathrm{dist}_K(x)$ is the Euclidean distance between $x$ and its $K$-th nearest neighbor, and $\mathbb{1}\{\cdot\} \in \{0, 1\}$ is the indicator function that evaluates the truth of the input condition.

Our clustering method, denoted by P-RWDKC($\alpha, \gamma, t, t_d$), is endowed with the parameters $\alpha \geq 0$, $\gamma \in [0, 1]$, and $t, t_d \geq 0$. The search grid used for each parameter is: $\alpha \in \{0, 0.1, ..., 1\}$, $t \in \{0, 1, ..., 100\}$, and $\gamma \in \{0, 0.1, ..., 1\}$. Finally, the diffusion time parameter $t_d \in \{2^0, ..., 2^J\}$, with $J = 15$, is estimated using Alg. 2. For each method, we apply $k$-means clustering over the obtained embeddings (we report the best score out of 100 restarts). We select for each method the optimal parameter values obtained through cross-validation over a grid search, yielding the closest partition to the ground truth. The obtained partitions are evaluated by the normalized mutual information (NMI) Strehl & Ghosh (2002), which is a popular supervised cluster evaluation index. NMI corresponds to a normalization of the mutual information between the predicted cluster assignments and the ground truth labels. This metric is symmetric and invariant to label permutations.

In the experiments we compare with the following methods:
• DSC $+ (\gamma)$ (Zhou et al., 2005) is a SC method on digraphs based on the Pagerank random walk (Page et al., 1999) endowed with the parameter $\gamma \in [0, 1)$.
• DI-SIM$_L(\tau)$ and DI-SIM$_R(\tau)$ (Rohe et al., 2016) are two variants that are based on the left and the right singular vectors, respectively, of a given regularized and normalized operator whose regularization is denoted by the parameter $\tau \geq 0$. We use cross-validation to search the optimal parameter with a grid search over $\tau \in \{1, 2, ..., 20\}$.
• SC-SYM$_1$ and SC-SYM$_2$ are SC variants (Von Luxburg, 2007) based on the unnormalized and the normalized graph Laplacians obtained from the symmetrization of the adjacency matrix $\mathbf{W}$.
• PIC($t_d$) (Lin & Cohen, 2010) is the clustering approach based on the power iteration of the random walk operator. We use the random walk operator derived from the original adjacency matrix $\mathbf{W}$ of the graph. The diffusion time $t_d$ is estimated using Alg. 2.
• RSC($\tau$) (Qin & Rohe, 2013; Zhang & Rohe, 2018) is the regularized SC proposed to deal with sparse graphs parametrized by $\tau \geq 0$. We use cross-validation with a grid search over $\tau \in \{1, 2, ..., 20\}$ to tune this parameter. The method applies only to undirected graphs. For digraphs, we had to symmetrize the adjacency matrix of the original digraph.

**Multi-scale synthetic Gaussians.** Here we test the our approach at revealing clusters at different scales thanks to the accurate estimation of the diffusion time. We generate *one instance* of a point cloud in $\mathbb{R}^2$, of $N = 300$ data points drawn independently from the following mixture of six Gaussian distributions $\sum_{i=1}^6 \alpha_i \mathcal{N}(\mu_i, \sigma_i^2 \boldsymbol{I})$ with weights $\alpha_i$ (note: $\sum_i \alpha_i = 1$). Specifically, $\sigma_i = 0.5, \alpha_i = 1/6, \forall i$ and $\mu_1 = (-3, -2), \mu_2 = (0, -2), \mu_3 = (-1, 1), \mu_4 = (4, -2), \mu_5 = (7, -2), \mu_6 = (5, 1)$. The resulting data exhibit a multi-scale structure, as they can be seen as either

**Table 1:** Clustering performance (NMI) on UCI datasets with optimal parameters in parentheses.

| DATASET | $N$ | $d$ | $k$ | SC-SYM$_1$ | SC-SYM$_2$ | DI-SIM$_L(\tau)$ | DI-SIM$_R(\tau)$ | DSC+$(\gamma)$ | PIC$(t_d)$ | P-RWDKC$(\alpha,\gamma,t,t_d)$ |
|---|---|---|---|---|---|---|---|---|---|---|
| IRIS | 150 | 4 | 3 | 80.58 | 80.58 | 74.98 (1) | 66.57 (1) | 68.63 (0.80) | 78.32 (32) | **90.11** (0.4,1,49,32) |
| GLASS | 214 | 9 | 6 | 38.59 | 38.92 | 38.95 (1) | 36.41 (1) | 39.72 (0.80) | 42.79 (128) | **44.39** (0.9,0.6,1,256) |
| WINE | 178 | 13 | 3 | 86.33 | 86.33 | 83.66 (1) | 85.62 (1) | **91.09** (0.80) | 86.33 (4) | 86.50 (0.7,0,21,32) |
| WBDC | 569 | 30 | 2 | 67.73 | 69.47 | 68.54 (2) | 53.43 (1) | 61.12 (0.10) | 64.77 (8) | **73.90** (0.9,0.3,100,2) |
| CONTROL CHART | 600 | 60 | 6 | 81.17 | 81.17 | 82.94 (1) | 77.72 (1) | 79.45 (0.90) | 82.79 (32) | **87.49** (0.8,1,75,32) |
| PARKINSON | 185 | 22 | 2 | 21.96 | 19.13 | 28.89 (1) | 27.36 (13) | 25.82 (0.95) | 28.89 (2) | **36.08** (1,0.3,52,2) |
| VERTEBRAL | 310 | 6 | 3 | 39.26 | 39.26 | 52.06 (2) | 41.76 (2) | 56.63 (0.80) | 49.13 (8) | **62.40** (0.3,0.8,48,4) |
| BREAST TISSUE | 106 | 9 | 6 | 54.03 | 54.43 | 54.04 (2) | 49.33 (2) | 51.64 (0.20) | 54.18 (32) | **60.43** (0.5,1,39,16) |
| SEEDS | 210 | 7 | 3 | 73.90 | 73.90 | 76.29 (1) | 73.06 (1) | 74.80 (0.80) | 70.79 (32) | **78.95** (1.0,0.4,84,8) |
| IMAGE SEG. | 2310 | 19 | 7 | 67.06 | 67.41 | 67.42 (1) | 64.77 (1) | 31.83 (0.99) | 69.58 ($2^{14}$) | **72.19** (0.8,1,80,256) |
| YEAST | 1484 | 8 | 10 | 30.58 | 31.11 | 31.37 (2) | 28.89 (1) | 27.50 (0.90) | 32.62 (16) | **34.05** (0.7,0.6,51,16) |
| AVERAGE | – | – | – | 58.29 | 58.34 | 59.92 | 54.77 | 56.37 | 60.01 | **66.04** |

containing the 6 original Gaussian clusters, or as having 2 clusters where each of them is made up of 3 smaller clusters. Figs. 1a and 1c show the ground truth data classes for the two scales. Figs. 1b and 1d, show the respective clustering results obtained by P-RWDKC with a vertex measure $\nu = \mathbf{1}$ and estimated diffusion times $t_{d_1} = 64$ and $t_{d_2} = 128$. The results are quite consistent with the ground truth of each case, hence provide evidence that our approach can reveal multi-scale clusters.

**Real-world data.** In this section, we conduct experiments on graphs created from high-dimensional real-world data. We show that P-RWDKC has superior performance compared to existing methods in nearly all tested cases. Note that, to further validate the efficiency of P-RWDKC, we run additional experiments on several real-world graphs, which are provided in Appendix A.2.

Here we report results on 11 benchmark datasets from the UCI repository (Dheeru & Karra Taniski-dou, 2017). We compare against DSC+, SC-SYM$_1$ and SC-SYM$_2$, DI-SIM, and PIC. Tab. 1 summarizes the comparative results based on NMI. In nearly all cases, the proposed P-RWDKC outperforms significantly the other methods on average. Our approach performs better than SC-SYM$_1$ and SC-SYM$_2$. Moreover, our approach performs better on average than DSC+. This allows us to state that P-RWDKC, associated with the suitable vertex measure from Eq. 9 brings indeed real added value to the clustering problem. Furthermore, P-RWDKC outperforms PIC. Consequently, the RWDK operator produces better graph embeddings than the original random walk operator defined the directed $K$-NN graphs because the parametrized random walk is irreducible compared to the random walk used in PIC that is not and thanks to the digraph information encoded into the vertex measure. We have to mention that for each dataset, the parameters reported in our approach are not necessarily unique as several combinations of parameters may yield the same clustering performance (see Sec. 5.1.)

## 7   CONCLUSION

We have proposed the *parametrized random walk diffusion kernel clustering* (P-RWDKC) that applies to both directed and undirected graphs. First, we introduced the parametrized random walk (P-RW) operator. We then show that the diffusion distance, is a Mahalanobis distance involving a special kernel matrix, called random walk diffusion kernel (RWDK), which is simply a power of the transition matrix (normalized by the vertex degrees). From this, we show that the RWDK is an alternative to the eigendecomposition step of the spectral clustering pipeline. We extend the diffusion geometry framework to digraphs by combining RWDK and P-RW. The P-RWDKC clustering algorithm stems from our analysis. Finally, we demonstrated empirically, with extensive experiments on several datasets, that P-RWDKC outperforms existing approaches for digraphs.

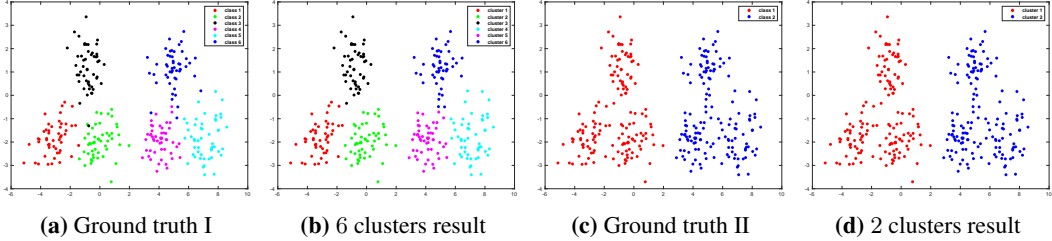

| **(a)** Ground truth I | **(b)** 6 clusters result | **(c)** Ground truth II | **(d)** 2 clusters result |
|---|---|---|---|

**Figure 1:** Comparison between the ground truth classes at different scales and the result of P-RWDKC on a synthetic toy-case. (a),(b) Small-scale: ground truth with 6 clusters and the P-RWDKC's result. (c),(d) Large-scale: ground truth with 2 clusters and the finding of P-RWDKC.

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

# A   APPENDIX

## A.1   PROOFS

### A.1.1   PROOF OF PROPOSITION 4.1

**Proposition A.1.** *Let $\mathcal{X}$ be a random walk on an undirected graph $\mathcal{G}$ with transition matrix $\mathbf{P}$ and ergodic distribution $\pi$. The transition matrix $\mathbf{P}$ admits the following eigendcomposition $\mathbf{P} = \mathbf{\Phi D}_\lambda \mathbf{\Psi}^\mathsf{T}$. The diffusion distance between vertices $i$ and $j$, $d_t^2(i,j)$ at a given time $t \in \mathbb{N}$ can be written as the following Mahalanobis distance*

$$d_t^2(i,j) = (\delta_i - \delta_j)^\mathsf{T} \mathbf{K}_t (\delta_i - \delta_j),$$

*where the similarity positive definite kernel matrix $\mathbf{K}_t$ is defined by*

$$\mathbf{K}_t = \mathbf{P}^{2t} \mathbf{D}_d^{-1}. \tag{11}$$

*Proof.*

$$
\begin{aligned}
d_t^2(i,j) &= \|p_t(i,*) - p_t(j,*)\|_{1/\pi}^2, \\
&= \|(\mathbf{P}^t)^\mathsf{T}(\delta_i - \delta_j)\|_{1/\pi}^2, \\
d_t^2(i,j) &= (\delta_i - \delta_j)^\mathsf{T} \mathbf{P}^t \mathbf{D}_\pi^{-1} (\mathbf{P}^t)^\mathsf{T} (\delta_i - \delta_j).
\end{aligned}
$$

By setting $\mathbf{K}_t = \mathbf{P}^t \mathbf{D}_\pi^{-1} (\mathbf{P}^t)^\mathsf{T}$, using the eigendecomposition of $\mathbf{P}$ and the fact that $\mathbf{P}$ is self-adjoint in $\ell^2(\mathcal{V}, \pi)$, we have

$$
\begin{aligned}
\mathbf{K}_t &= \mathbf{P}^t \mathbf{D}_\pi^{-1} (\mathbf{P}^t)^\mathsf{T} \\
&= \mathbf{\Phi D}_\lambda^t \mathbf{\Psi}^\mathsf{T} \mathbf{D}_\pi^{-1} \mathbf{\Psi D}_\lambda^t \mathbf{\Phi}^\mathsf{T}, \\
&= \mathbf{\Phi D}_\lambda^{2t} \mathbf{\Phi}^\mathsf{T} \ (\mathbf{\Psi}^\mathsf{T} \mathbf{D}_\pi^{-1} \mathbf{\Psi} = \mathbf{I}), \\
&= \mathbf{\Phi D}_\lambda^{2t} \mathbf{\Psi}^\mathsf{T} \mathbf{D}_d^{-1} \ (\mathbf{\Phi} = \mathbf{D}_d^{-1} \mathbf{\Psi}), \\
\mathbf{K}_t &= \mathbf{P}^{2t} \mathbf{D}_d^{-1}.
\end{aligned}
$$

$\square$

### A.1.2   SUPPLEMENTARY PROOFS

**Proposition A.2.** $\mathbf{P}_{(\nu)}$ *is a transition matrix and reversible.*

*Proof.* We have the following equality

$$\mathbf{P}_{(\nu)} = (\mathbf{I} + \mathbf{D}_\nu^{-1} \mathbf{D}_\xi)^{-1} (\mathbf{P} + \mathbf{D}_\nu^{-1} \mathbf{P}^\mathsf{T} \mathbf{D}_\nu) = (\mathbf{D}_\nu + \mathbf{D}_\xi)^{-1} (\mathbf{D}_\nu \mathbf{P} + \mathbf{P}^\mathsf{T} \mathbf{D}_\nu).$$

As a result, we need to show that $(\mathbf{D}_\nu + \mathbf{D}_\xi)^{-1} (\mathbf{D}_\nu \mathbf{P} + \mathbf{P}^\mathsf{T} \mathbf{D}_\nu)$ is a transition matrix, i.e. we show that

$$\sum_{j=1}^{|\mathcal{V}|} \mathbf{P}_{(\nu),ij} = 1, \quad \forall i \in \mathcal{V}$$

$$
\begin{aligned}
\sum_{j=1}^{|\mathcal{V}|} \mathbf{P}_{(\nu),ij} &= \sum_{j=1}^{|\mathcal{V}|} \left( (\mathbf{D}_\nu + \mathbf{D}_\xi)^{-1} (\mathbf{D}_\nu \mathbf{P} + \mathbf{P}^\mathsf{T} \mathbf{D}_\nu) \right)_{ij} \\
&= \sum_{j=1}^{|\mathcal{V}|} \left( \sum_{k=1}^{|\mathcal{V}|} (\mathbf{D}_\nu + \mathbf{D}_\xi)_{ik}^{-1} (\mathbf{D}_\nu \mathbf{P} + \mathbf{P}^\mathsf{T} \mathbf{D}_\nu)_{kj} \right) \\
\sum_{j=1}^{|\mathcal{V}|} \mathbf{P}_{(\nu),ij} &= (\mathbf{D}_\nu + \mathbf{D}_\xi)_{ii}^{-1} \sum_{j=1}^{|\mathcal{V}|} (\mathbf{D}_\nu \mathbf{P} + \mathbf{P}^\mathsf{T} \mathbf{D}_\nu)_{ij} \tag{12}
\end{aligned}
$$

$$\sum_{j=1}^{|\mathcal{V}|}(\mathbf{D}_\nu\mathbf{P} + \mathbf{P}^\mathsf{T}\mathbf{D}_\nu)_{ij} = \sum_{j=1}^{|\mathcal{V}|}\sum_{k=1}^{|\mathcal{V}|}(\mathbf{D}_\nu)_{ik}\mathbf{P}_{kj} + \sum_{j=1}^{|\mathcal{V}|}\sum_{k=1}^{|\mathcal{V}|}(\mathbf{P}^\mathsf{T})_{ik}(\mathbf{D}_\nu)_{kj}$$

$$= \sum_{j=1}^{|\mathcal{V}|}(\mathbf{D}_\nu)_{ii}\mathbf{P}_{ij} + \sum_{j=1}^{|\mathcal{V}|}(\mathbf{P}^\mathsf{T})_{ij}(\mathbf{D}_\nu)_{jj}$$

$$= \sum_{j=1}^{|\mathcal{V}|}\nu(i)p(i,j) + \sum_{j=1}^{|\mathcal{V}|}\nu(j)p(j,i)$$

$$\sum_{j=1}^{|\mathcal{V}|}(\mathbf{D}_\nu\mathbf{P} + \mathbf{P}^\mathsf{T}\mathbf{D}_\nu)_{ij} = \nu(i) + \xi(i), \quad \forall\, i \in \mathcal{V}$$

Using Eq. 12, we finally obtain $\sum_{j=1}^{|\mathcal{V}|}\mathbf{P}_{(\nu),ij} = 1$. $\qquad\square$

## A.2 EXPERIMENTS ON REAL-WORLD GRAPH BENCHMARKS

Most of the real-world graphs are sparse with heterogeneous degrees with a high variance of the degree. As a result, spectral clustering mostly fails for these graphs Zhang & Rohe (2018); Rohe et al. (2011); Dall'Amico et al. (2020). To avoid numerical issues due to the high variance degree, we proposed a slight modification of the parametrized random walk operator (transition matrix)

$$\mathbf{P}_{(\nu)} = (\mathbf{D}_{\tilde{\nu}} + \mathbf{D}_\xi)^{-1}(\mathbf{D}_\nu\mathbf{W} + \mathbf{W}^\mathsf{T}\mathbf{D}_\nu) \tag{13}$$

with $\xi = \nu^\mathsf{T}\mathbf{W}$ and the vertex measure $\tilde{\nu} = [\tilde{\nu}_1, ..., \tilde{\nu}_N]^\mathsf{T}$ defined by $\tilde{\nu}_i = \nu_i d_i^{\text{out}}, \forall i \in \mathcal{V}$.

We report the results of experiments that evaluate the performance of the proposed P-RWDKC method on 4 real-world networks. Among these, 2 are directed (Political blogs, Cora) and 2 are undirected (Karate Club, College Football). For all networks, the number of clusters is considered known. Moreover, when necessary, clustering is performed on the largest connected component of the graph. We compare against SC-SYM$_1$, RSC, and PIC.

Tab. 2 summarizes the comparative results according to the NMI index. In all cases, the proposed P-RWDKC outperforms the other methods, and when looking at the overall average performance the difference is significant. Our approach performs significantly better than SC-SYM$_1$. This is caused by a mixed effect of the symmetrization of the digraph as well as the hard truncation phenomenon discussed in Sec. 4.2. RSC is competitive against our method. Nevertheless, the adjacency matrix involved in RSC is dense and clearly modified, which can have a deleterious impact compared to our method (e.g. see the results on Cora).

**Table 2:** Clustering performance (NMI) on real-world datasets with optimal parameters in parentheses.

| DATASET | $N$ | $k$ | SC-SYM$_1$ | RSC($\tau$) | PIC($t_d$) | P-RWDKC($\alpha,\gamma,t,t_d$) |
|---|---|---|---|---|---|---|
| POLITICAL BLOGS (ADAMIC & GLANCE, 2005) | 1222 | 2 | 1.74 | 73.25 (0.2) | 57.37 (8) | **75.53** (0.2,1,17,32) |
| CORA (BOJCHEVSKI & GÜNNEMANN, 2017) | 2485 | 7 | 17.10 | 36.07 (2.5) | 7.86 (4) | **52.03** (0,0.9,72,8) |
| COLLEGE FOOTBALL (GIRVAN & NEWMAN, 2002) | 115 | 12 | 29.97 | 30.79 (2.9) | 29.98 (2) | **31.30** (0.7,0,17,2) |
| KARATE CLUB (ZACHARY, 1977) | 34 | 2 | 73.24 | **83.72** (1.7) | **83.72** (1) | **83.72** (0,0,1,1) |
| AVERAGE | – | – | 30.51 | 55.96 | 49.23 | **60.65** |

## A.3 ALTERNATIVE FOR THE DESIGN OF THE VERTEX MEASURE

In Sec. 5.1 we presented a design for the vertex measure to be used by P-RWDKC. Here we discuss an alternative design. As before, the vertex measure can be parametrized by three parameters ($t \in \mathbb{N}$, $\gamma \in [0, 1]$, $\alpha \in \mathbb{R}$) and is formally given by:

$$\nu_{(t,\gamma)}^\alpha(i) = \left(\tfrac{1}{N}\mathbf{1}_{N\times 1}^\mathsf{T}\mathbf{P}_\gamma^t\delta_i\right)^\alpha, \tag{14}$$

where $\mathbf{1}_{N\times 1}$ is the all-ones vector , $\delta_i \in \{0,1\}^{N\times 1}$ is the vector output of the Kronecker delta function at $i \in \mathcal{V}$, and

$$\mathbf{P}_\gamma = \mathbf{D}_\gamma^{-1}\mathbf{W}_\gamma, \quad \mathbf{D}_\gamma = \text{diag}(\mathbf{W}_\gamma\mathbf{1}), \tag{15}$$

**Table 3:** Clustering performance (NMI) on UCI datasets with optimal parameters in parentheses.

| DATASET | $N$ | $d$ | $k$ | SC-SYM$_1$ | SC-SYM$_2$ | DI-SIM$_L(\tau)$ | DI-SIM$_R(\tau)$ | DSC+$(\gamma)$ | PIC$(t_d)$ | P-RWDK$(\alpha, t_d)$ |
|---|---|---|---|---|---|---|---|---|---|---|
| IRIS | 150 | 4 | 3 | 80.58 | 80.58 | 74.98 (1) | 66.57 (1) | 68.63 (0.80) | 78.32 (32) | **90.11** (0.6,4) |
| GLASS | 214 | 9 | 6 | 38.59 | 38.92 | 38.95 (1) | 36.41 (1) | 39.72 (0.80) | 42.79 (128) | **42.89** (0.5,256) |
| WINE | 178 | 13 | 3 | 86.33 | 86.33 | 83.66 (1) | 85.62 (1) | **91.09** (0.80) | 86.33 (4) | 84.73 (0,32) |
| WBDC | 569 | 30 | 2 | 67.73 | 69.47 | 68.54 (2) | 53.43 (1) | 61.12 (0.10) | 64.77 (8) | **71.32** (1,2) |
| CONTROL CHART | 600 | 60 | 6 | 81.17 | 81.17 | 82.94 (1) | 77.72 (1) | 79.45 (0.90) | 82.79 (32) | **84.22** (0.6,32) |
| PARKINSON | 185 | 22 | 2 | 21.96 | 19.13 | 28.89 (1) | 27.36 (13) | 25.82 (0.95) | 28.89 (2) | **36.08** (0.5,4) |
| VERTEBRAL | 310 | 6 | 3 | 39.26 | 39.26 | 52.06 (2) | 41.76 (2) | 56.63 (0.80) | 49.13 (8) | **59.57** (0.1,4) |
| BREAST TISSUE | 106 | 9 | 6 | 54.03 | 54.43 | 54.04 (2) | 49.33 (2) | 51.64 (0.20) | 54.18 (32) | **55.29** (0.6,16) |
| SEEDS | 210 | 7 | 3 | 73.90 | 73.90 | 74.89 (1) | 73.06 (1) | 74.80 (0.80) | 70.79 (32) | **75.52** (0,8) |
| IMAGE SEG. | 2310 | 19 | 7 | 67.06 | 67.41 | 67.42 (1) | 64.77 (1) | 31.83 (0.99) | 69.58 ($2^{14}$) | **70.70** (0.5,64) |
| YEAST | 1484 | 8 | 10 | 30.58 | 31.11 | 31.37 (2) | 28.89 (1) | 27.50 (0.90) | 32.62 (16) | **33.83** (0.1,16) |
| AVERAGE | – | – | – | 58.29 | 58.34 | 59.92 | 54.77 | 56.37 | 60.01 | **63.92** |

where $\mathbf{W}_\gamma$ is defined by

$$\mathbf{W}_\gamma = \gamma \mathbf{W} + (1 - \gamma)\mathbf{W}^\mathsf{T}, \quad \gamma \in [0, 1]. \tag{16}$$

The random walk iteration parameter $t$ controls the random walk diffusion, $\gamma$ controls the influence between the original adjacency $\mathbf{W}$ (forward information) and its transpose $\mathbf{W}^\mathsf{T}$ (backward information), and $\alpha$ controls the re-weighting of the vertex measure.

### A.3.1 VERTEX MEASURE WHEN $\gamma = 1/2$ AND $t \to \infty$

In the setting where, $\gamma = \frac{1}{2}$ and $t \to \infty$, we are able to characterize explicitly the vertex measure. Indeed, $\mathbf{W}_\gamma$ becomes symmetric, the associated random walk is thus ergodic, $\lim_{t\to\infty} \delta_i^\mathsf{T} \mathbf{P}_{1/2}^t = \pi_{1/2}, \forall i$ and the parametrized vertex measure defined in Eq. 14 becomes $\nu_{(t,\gamma)}^\alpha(i) = \left(\pi_{1/2}(i)\right)^\alpha$ for any vertex $i$.

### A.3.2 ADDITIONAL EXPERIMENTS

In this section, we demonstrate the effectiveness of our approach based on the same setting described in Sec. 6, using the vertex measure in Sec. A.3.1. Tabs. 3 and 5 summarize the comparative results based on NMI. Similarly to the results from Sec. 6, we observe that the proposed P-RWDKC outperforms significantly the other methods in nearly all cases. The P-RWDKC, associated with the vertex measure defined in Sec. A.3.1 stays competitive against P-RWDKC from Sec. 6 with less degree of freedom. The main takeaway to consider is that the construction of this kernel and the design of the vertex measures are the key elements to creating relevant embedding for clustering graphs.

To further evaluate the P-RWDKC framework, instead of using the ground truth and cross-validation, here we choose the parameter values that optimize the Calinski-Harabasz (CH). We first compute the set of candidate partitions, one for each $(\alpha, t_d)$ combination in the considered parameter grid. Then we select as best the model with the highest CH and compute its NMI. We operate in the same way for the methods that have parameters (i.e. all but SC-SYM$_1$, SC-SYM$_2$, and PIC whose results are the same as in Tabs. 3 and 5. The comparative results are shown in Tabs. 4 and 6. Notice that P-RWDKC outperforms significantly the other methods in nearly all cases (see also the average performance in the last row of the table.). Compared to Tab. 3 here the NMI of P-RWDKC stays just a little lower. This indicates that the unsupervised tuning of the model parameters offers comparable graph partition quality to the previous case where we applied cross-validation using the ground truth.

**Table 4:** Clustering performance (NMI) on UCI datasets with estimated parameters (shown in parentheses) according to the CH (or DCH) index.

| DATASET | $N$ | $d$ | $k$ | SC-SYM$_1$ | SC-SYM$_2$ | DI-SIM$_L(\tau)$ | DI-SIM$_R(\tau)$ | DSC+$(\gamma)$ | PIC$(t_d)$ | P-RWDK$(\alpha, t_d)$ |
|---|---|---|---|---|---|---|---|---|---|---|
| IRIS | 150 | 4 | 3 | 80.58 | 80.58 | 74.98 (1) | 66.57 (1) | 68.63 (0.80) | 78.32 (32) | **90.11** (0.6,4) |
| GLASS | 214 | 9 | 6 | 38.59 | 38.92 | 37.39 (2) | 35.87 (1) | 36.58 (0.80) | 42.79 (128) | **42.13** (0.3,256) |
| WINE | 178 | 13 | 3 | 86.33 | 86.33 | 83.66 (1) | 85.62 (1) | **91.09** (0.80) | 86.33 (4) | 84.73 (0.9,32) |
| WBDC | 569 | 30 | 2 | 67.73 | 69.47 | 68.54 (2) | 53.43 (1) | 61.12 (0.10) | 64.77 (8) | **70.24** (0.5,2) |
| CONTROL CHART | 600 | 60 | 6 | 81.17 | 81.17 | 82.94 (2) | 77.44 (1) | 79.45 (0.90) | 82.79 (32) | **83.19** (0.2,32) |
| PARKINSON | 185 | 22 | 2 | 21.96 | 19.13 | 28.89 (1) | 27.36 (13) | 22.97 (0.95) | 28.89 (2) | **36.08** (0.7,4) |
| VERTEBRAL | 310 | 6 | 3 | 39.26 | 39.26 | 45.89 (2) | 39.62 (1) | 54.24 (0.80) | 49.13 (8) | **59.34** (0.3,4) |
| BREAST TISSUE | 106 | 9 | 6 | 54.03 | 54.43 | 54.04 (2) | 49.27 (2) | 51.64 (0.20) | 54.18 (32) | **54.90** (0.2,16) |
| SEEDS | 210 | 7 | 3 | 73.90 | 73.90 | **74.89** (1) | 73.06 (1) | 74.80 (0.80) | 70.79 (32) | **74.89** (0.1,8) |
| IMAGE SEG. | 2310 | 19 | 7 | 67.06 | 67.41 | 67.42 (1) | 64.77 (1) | 31.46 (0.99) | 69.58 ($2^{14}$) | **68.79** (0.1,64) |
| YEAST | 1484 | 8 | 10 | 30.58 | 31.11 | 31.22 (1) | 28.89 (1) | 27.47 (0.90) | 32.62 (16) | **33.60** (0,16) |
| AVERAGE | – | – | – | 58.29 | 58.34 | 58.82 | 54.57 | 51.95 | 60.01 | **63.45** |

**Table 5:** Clustering performance (NMI) on real-world datasets with optimal parameters in parentheses.

| DATASET | $N$ | $k$ | SC-SYM$_1$ | RSC$(\tau)$ | PIC$(t_d)$ | P-RWDKC$(\alpha, t_d)$ |
|---|---|---|---|---|---|---|
| POLITICAL BLOGS (ADAMIC & GLANCE, 2005) | 1222 | 2 | 1.74 | 73.25 (0.2) | 57.37 (8) | **74.02** (0.1,32) |
| CORA (BOJCHEVSKI & GÜNNEMANN, 2017) | 2485 | 7 | 17.10 | 36.07 (2.5) | 7.86 (4) | **49.71** (0.2,16) |
| COLLEGE FOOTBALL (GIRVAN & NEWMAN, 2002) | 115 | 12 | 29.97 | 30.79 (2.9) | 29.98 (2) | **30.79** (0.9,2) |
| KARATE CLUB (ZACHARY, 1977) | 34 | 2 | 73.24 | **83.72** (1.7) | **83.72** (1) | **83.72** (0,1) |

## A.4 THEORETICAL BACKGROUND

### A.4.1 BACKGROUND ON GENERALIZED GRAPH LAPLACIANS

The generalized graph Laplacians proposed in Sevi et al. (2022) are the generalization of the usual graph Laplacians found in the literature Chung & Graham (1997); Chung (2005). These operators stem directly from a functional termed as *generalized Dirichlet energy* (GDE) expressed in the following definition.

**Definition A.1.** *Let $\mathcal{X}$ be a random walk on a digraph $\mathcal{G}$, with transition matrix $\mathbf{P}$. Let $\nu$ be an arbitrary positive vertex measure on $\mathcal{G}$. The generalized Dirichlet energy of a graph function $f$ is defined by*

$$\mathcal{D}_{\nu,\mathbf{P}}^2(f) = \sum_{x,y \in \mathcal{V}} \nu(x)p(x,y)|f(x) - f(y)|^2. \tag{17}$$

This functional extends the well-known notion of Dirichlet energy Montenegro et al. (2006) which is originally restricted to any ergodic random walk associated with its transition matrix and its stationary measure. In particular, the GDE is defined with respect to any positive regularizing measure and any Markov transition matrix without the requirement of any specific property. As mentioned earlier, the generalized graph Laplacians stems from the GDE by the following relation

$$\mathcal{D}_{\nu,\mathbf{P}}^2(f) = \langle f, \mathbf{L}_{\text{RW}}(\nu)f \rangle_{\nu+\xi} = \langle f, \mathbf{L}(\nu)f \rangle.,$$

which means that the quadratic form of $\mathcal{D}_{\nu,\mathbf{P}}^2(f)$ involves the unnormalized generalized graph Laplacian. It is important to mention that the GDE stems directly from the random walk point of view of the graph partitioning proposed in Sevi et al. (2022).

### A.4.2 BACKGROUND ON DIFFUSION GEOMETRY

The seminal work by (Coifman et al., 2005; Coifman & Lafon, 2006) introduced the *diffusion geometry framework*, which uses diffusion processes as a basic tool to find meaningful geometric descriptions for data. This general framework leads to efficient multi-scale analysis of datasets for which we have a Heisenberg location principle relating localization in data to localization in the spectrum. The framework can provide different geometric representations of the dataset by iterating the Markov transition matrix, which is equivalent to running forward the random walk. The key element of the diffusion geometry is the diffusion distance Coifman & Lafon (2006); Pons & Latapy (2005); Coifman et al. (2005) which captures the structure encoded in $\mathbf{P}$ as a data-dependent distance metric between points and whose the original definition is the following.

**Table 6:** Clustering performance (NMI) on real-world datasets with estimated parameters (shown in parentheses) according to the CH (or DCH) index.

| DATASET | $N$ | $k$ | SC-SYM$_1$ | RSC($\tau$) | PIC($t_d$) | P-RWDKC($\alpha, t_d$) |
|---|---|---|---|---|---|---|
| POLITICAL BLOGS (ADAMIC & GLANCE, 2005) | 1222 | 2 | 1.74 | 73.25 (0.2) | 57.37 (8) | **73.93** (0,32) |
| CORA (BOJCHEVSKI & GÜNNEMANN, 2017) | 2485 | 7 | 17.10 | 30.65 (0.5) | 7.86 (4) | **49.71** (0.2,16) |
| COLLEGE FOOTBALL (GIRVAN & NEWMAN, 2002) | 115 | 12 | 29.97 | 29.84 (2.9) | 29.98 (2) | **30.30** (0.9,2) |
| KARATE CLUB (ZACHARY, 1977) | 34 | 2 | 73.24 | **83.72** (1.7) | **83.72** (1) | **83.72** (0,1) |

**Definition A.2.** *Diffusion distance. Let $\mathbf{P}$ be the transition matrix of a reversible random walk on an undirected graph $\mathcal{G}$, with an ergodic distribution $\pi$. Given a graph function $f$, $\|f\|_{1/\pi}^2$ is the $\ell^2$-norm induced by the measure $1/\pi$ defined by $\|f\|_{1/\pi}^2 = \langle f, \mathbf{D}_\pi^{-1} f \rangle$. The diffusion distance at time $t \in \mathbb{N}$ between the vertices $i$ and $j$ is defined by:*

$$d_t^2(i,j) = \|p_t(i,*) - p_t(j,*)\|_{1/\pi}^2.$$

Notably, diffusion distances are data-dependent, allowing the detection of nonlinear structure in data at different scales Coifman et al. (2005). The diffusion distance at time $t$ can be considered as the Euclidean distance between rows of $\mathbf{P}^t$, potentially weighted by a measure (in the original definition by $1/\pi$) which takes into account the (empirical) local density of the points. For a finite time $t$, if each cluster in a cluster of $\mathcal{G}$ is highly-connected, and well-separated from other clusters, hen $p_t(i,*)$ will be nearly equal to $p_t(j,*)$ for any pair of points $i$ and $j$ in the same cluster, implying a low diffusion distance between points within the same cluster. Conversely, if $i$ and $j$ are in distinct clusters, $p_t(i,*)$ is expected to be very different from $p_t(j,*)$ . Since then, some variants of the diffusion distance notion have been proposed Goldberg & Kim (2010; 2012).

We now introduce the notion of diffusion map as a main concept of the diffusion geometry framework.

**Definition A.3.** *Diffusion map. Let $\mathcal{X}$ be a random walk on an undirected graph $\mathcal{G} = (\mathcal{V}, \mathcal{E}), |\mathcal{V}| = N$ with transition matrix $\mathbf{P}$ and ergodic distribution $\pi$. The transition matrix $\mathbf{P}$ admits the following eigendecomposition $\mathbf{P} = \mathbf{\Phi}\mathbf{D}_\lambda\mathbf{\Phi}^{-1}$ where $\mathbf{\Phi} = [\phi_1, \phi_2, \cdots, \phi_n]$ is the eigenbasis and $\mathbf{D}_\lambda = \mathrm{diag}(\lambda)$ is the diagonal matrix of eigenvalues. The diffusion map at time $t$ is defined by*

$$\Psi_t(i) = (\lambda_1^t \phi_1(i), \lambda_2^t \phi_1(i), \cdots, \lambda_n^t \phi_n(i)).$$

Diffusion maps and diffusion distances are related by the following relation Coifman & Lafon (2006)

$$d_t^2(i,j) = \|\Psi_t(i) - \Psi_t(j)\|^2,$$

which means that the diffusion map $\Psi_t$ embeds the data into a Euclidean space in which the Euclidean distance is equal to the diffusion distance $d_t^2$. We note that this relationship only exists if the diffusion distance is the original form proposed in Coifman & Lafon (2006), i.e. if the random walk with transition matrix $\mathbf{P}$ is reversible and if the Euclidean distance is weighted by $1/\pi$.

