# OpenReview forum: "Clustering for directed graphs using parametrized random walk diffusion kernels"
_ICLR.cc/2023/Conference — Submitted to ICLR 2023_

### Official Review · Reviewer_xrEQ · 2022-10-24

**Confidence:** 4
**Correctness:** 2
**Technical Novelty And Significance:** 2
**Empirical Novelty And Significance:** 2
**Recommendation:** 5

**Clarity, Quality, Novelty And Reproducibility:**

The exposition is general clear, and the main ideas are easy to follow. The novelty aspect is limited in my view as outlined in more detail in the “weaknesses” section. With respect to quality, I would say that the proposed method is promising as seen from the experiment results on real data. However, I am not sure if the method has been evaluated thoroughly from an empirical point of view, and there is also no theoretical analysis for a random generative model. I elaborate on these points in the “weaknesses” section.

**Strength And Weaknesses:**

Pros:  The paper has the following strengths.

•  The paper is well written overall, and the main ideas of the approach are outlined cleanly.

•  The experiment results on real data are promising as the proposed algorithm typically leads to better NMI scores than some of the existing approaches.

Cons: The paper has the following weaknesses.

• I feel that the technical novelty in the paper is quite limited. The parametrized random walk operator $P_{(\nu)}$ follows directly from the “generalized random walk Laplacian” operator $L_{RW, (\nu)}$ of (Sevi et al.,2022) since $P_{(\nu)} = I - L_{RW, (\nu)}$. This relation is currently stated in the form of Proposition 3.1, but this is a bit strange to me as there is nothing to prove here. The similarity kernel operator $K_{t,\nu}$ is obtained by taking the power of $P_{(\nu)}$ and subsequently normalizing it. However, this is quite similar to clustering approaches for undirected graphs which perform spectral clustering on the “the power of the adjacency matrix followed by an entry-wise truncation” (See for e.g. [1]). In essence, it is well-known that pre-processing a graph matrix (such as the adjacency) by taking a suitably high power of it (the exponent can be thought of as a regularizer term) leads to more robust clustering performance. So, I do not see new insight with regards to the proposed approach.

• The literature review is incomplete as there are many missing references for clustering directed graphs, see for e.g. [2,3,4,5] and reference therein. Therefore, I am not sure if the experiments are actually comparing with the state of the art for this problem. Additionally, I think it would have been beneficial to have more extensive experiments for synthetic examples – for e.g., on random generative models such as the directed Stochastic Block Model (SBM) (see [2]). Since the ground-truth is known, this would provide a clean framework for comparing the performance of the algorithms across different types of inputs (varying community sizes, noise levels, number of clusters etc.). The experiment results at the moment are incomplete in this sense.

• There also exist theoretical results for clustering directed graphs under a directed SBM generative model (e,g, [2]). In that sense, I believe it would have been interesting to provide some theoretical justification for the proposed approach on such a generative model.

Further comments:

• In Proposition 4.1, the notation $D_{\lambda}, D_d$ has not been defined, I think.

• As mentioned above, Proposition 3.1 should really be a Definition since there is nothing to prove here.

• The choice of the vertex measure $\nu$ and the diffusion time $t^*$ are based on heuristics at the moment. I think it is natural to run some experiments on synthetic examples generated by a directed SBM, for different choices of the tuning parameters (e.g., through a grid search), and then to check whether the best "global" value is close to that returned by the heuristic (or maybe just the corresponding NMI values could compared).

References:

[1] Abbe et al., Graph Powering and Spectral Robustness, SIAM J. MATH. DATA SCI, Vol. 2, No. 1, pp. 132–157.

[2] Cucuringu et al., Hermitian matrices for clustering directed graphs: insights and applications, AISTATS, PMLR 108:983-992, 2020.

[3] Laenen and Sun, Higher-order spectral clustering of directed graphs. In NIPS'20, 941–951.

[4] Gong et al., Directed network Laplacians and random graph models, R. Soc. open sci., 2021.

[5] Hayashi et al., Skew-Symmetric Adjacency Matrices for Clustering Directed Graphs, arxiv: 2203.01388, 2022


**Summary Of The Paper:**

The paper proposes a framework for clustering, namely the Parametrized Random Walk Diffusion Kernel Clustering (P-RWDKC), for clustering directed graphs. This is based on a “parametrized random walk operator” $P_{(\nu)}$ that is derived from the “generalized random walk Laplacian” operator of (Sevi et al.,2022), and using which a similarity kernel matrix $K_{t,\nu}$ is derived.  The matrix $K_{t,\nu}$ captures the pairwise diffusion distances between the vertices. The idea is to consider the rows of $ K_{t,\nu}$ as the embedding of the vertices, and so k-means clustering can be applied on the rows to estimate the underlying clustering. The proposed approach is tested on a synthetic example for clustering a mixture of Gaussians and also on several real datasets. The results on real data indicate that the proposed approach typically performs better than some other existing approaches.

**Summary Of The Review:**

The proposed approach seems to show promising results for clustering real-world directed graphs. However, I believe the technical novelty is a bit limited. The literature review and comparisons with related methods is also incomplete. More experiments on synthetic random generative models (such as directed SBM) are needed to understand the usefulness of the proposed approach as compared to the state-of-the-art methods. I have elaborated on these points in the "Weaknesses" section.

---

### Official Review · Reviewer_rHPo · 2022-10-25

**Confidence:** 3
**Correctness:** 4
**Technical Novelty And Significance:** 3
**Empirical Novelty And Significance:** 4
**Recommendation:** 6

**Clarity, Quality, Novelty And Reproducibility:**

The paper is fairly clear. The main ideas are described and motivated very nicely.

**Strength And Weaknesses:**

The theoretical strength of this work lies in realizing that we can define a (somewhat) natural diffusion kernel for directed graphs which is obtained by combining a "parameterized random walk operator" together with the notion of diffusion distance on undirected graphs. Though, at the same time, the paper does not present any algorithmic application which can benefit from this primitive. Having *not thought* much about it, I am not yet able to clearly compare the quality of clusters obtained from this procedure with the quality of clusters you would get from ACL algorithm which considers page rank walk on directed graphs.

As for the experimental side, I am fairly convinced (thanks to the included tables) that the algorithm presented in the paper does against its competitors. Though, I would request the authors to add a one-liner explaining what the NMI index

**Summary Of The Paper:**

The paper under review studies the clustering problem on directed graphs using random walks. First, let us recall the clustering problem on undirected graphs using random walks. The setup here is rather well-studied. You just perform a suitably long walk at a random vertex within the cluster. Using spectral methods, you recover a good approximation to the cluster by running the k-means-like algorithm on the Feidler embedding of the vertices. In the directed land, the spectral analysis immediately falls apart. But this is the least of our problems. A more serious problem is that the random walks on directed graphs may no longer be irreducible. To address this challenge, this paper develops a "random-walk diffusion-kernel-based" clustering approach.

Inspired by ideas from diffusion geometry, the paper defines a notion of parameterized diffusion distance at time t between every pair of vertices in the directed graph G. This is defined to be the Euclidean length of the difference in t-step probability vectors from u and v (in l_2(\pi_v)), Here, $\pi$ is the stationary distribution of parameterized diffusions on G (where parameterized diffusions are deliberately defined to have a stationary distribution). The clustering algorithm proceeds by running a k-means procedure on the spectral embedding found using the properties of this novel diffusion operator.

**Summary Of The Review:**

Overall, my feeling about this work is positive. Though, since the paper does not provide more theoretical details on the quality of the clusters returned by the novel diffusion process the paper considers, I am not able to fully recommend an accept verdict.

---

### Official Review · Reviewer_kddm · 2022-10-25

**Confidence:** 3
**Correctness:** 4
**Technical Novelty And Significance:** 2
**Empirical Novelty And Significance:** 3
**Recommendation:** 6

**Clarity, Quality, Novelty And Reproducibility:**

Minor remarks:
- In proposition 4.1, what is the use of specifying the eigendecomposition of P ? It is used in the proof, but not the statement or subsequent analysis.
- The notation $D_x$ is used throughout the paper to denote $diag(x)$, but it is inconsistently mentioned (e.g. it is in Definition 2.1 but not Proposition 4). I think it would be best to mention it in the preliminary section only.
- I don't understand the choice of presentation for $P_{\nu}$; it would probably be simpler to define $P$ first, and $L = I - P$ if needed. The rest of the laplacians add some confusion so early in the paper; they can maybe be moved so section 4.2. I also found the expression $P = D_{\nu + \xi}^{-1}(D_\nu P + P^\top D_\nu)$ much more intuitive, but it is only mentioned in the appendix.
- In Definition 4.1, the norm $\lVert \cdot \rVert_{1/\pi}$ is not standard, and should be defined
- there is a sentence in orange after Eq. 9
- in Appendix A.3.1, isn't it obvious that the measure only depends on $\alpha$, since you have set two of the three parameters ?


**Strength And Weaknesses:**

The paper is quite clear and enjoyable to read, and the presented algorithms are intuitive and clearly motivated. The main procedure seems to have a better performance than SOTA algorithms, and the example setting is interestingly chosen to show the possibilities of herarchical clustering.

My main concern is on the novelty of the paper: although interesting, this is simply a new embedding for k-means clustering, and as (nicely) explained in the paper it fits into the framework of other clustering algorithms. It is also unclear whether the benefits come from the choice of $P$, or the use of $K$ instead of spectral clustering.

**Summary Of The Paper:**

This paper concerns the clustering of directed graphs, using random walk diffusion. The main hurdle is that the digraphs considered may not be strongly connected, so the classical theory of irreducible random walks does not apply. To address this problem, the authors suggest to use a different random walk operator
$$P = D_{\nu + \xi}^{-1}(D_\nu P + P^\top D_\nu)$$
where $\nu$ can be chosen almost arbitrarily. This random walk is reversible and irreducible, hence the main hurdle is lifted. The authors also present a parametrizable family of choices for $\nu$, that are based on the original random walk matrix $P$.

Another aspect of the paper is information geometry. The authors show that a particular choice of distance for diffusion walks can be represented as a so-called Mahalanobis distance, with a given kernel $K$. This kernel can thus be used as an embedding for clustering. Additionally, they show how to measure the clustering of $P$ in a self-supervised way, and provide a stopping criterion based on this measure.

**Summary Of The Review:**

see above

---

### Official Review · Reviewer_pmG1 · 2022-10-25

**Confidence:** 3
**Correctness:** 4
**Technical Novelty And Significance:** 3
**Empirical Novelty And Significance:** 3
**Recommendation:** 6

**Clarity, Quality, Novelty And Reproducibility:**

Clarity:  Clear in writing and presenting. This paper provides a good amount of background for understanding their method. There are a few improvements that could be added (see 1st Weakness) to streamline the paper and make it more accessible to read.

Quality: Overall in good quality. The paper aims to address the directed graph clustering problem without losing the edge direction information, which is a very interesting and challenging topic, and also very timely research topic at the moment. The authors provide both theoretical understanding of the algorithms, and augment this with an extensive set of experiments and intuition, to demonstrate the merit and benefits of their design. For improvement see 2nd and 3rd Weakness.

Novelty: Very novel formulation.

Reproducibility: The authors provide a clear experiment setup and detailed explanations on parameter choosing for different methods.

**Strength And Weaknesses:**

 Strengths:
- The proposed clustering algorithm is symmetrization free, which avoids loss of the edge direction information. This has been the typical approach in the directed graph literature, and only very recently methods have been proposed which bypass this step
- The authors provide theoretical background for understanding and motivating their algorithm construction.
- This approach is simple, efficient and performs very well in experiments.

Weakness:
- Some part of the writing is not very well detailed:
On one hand, some important explanations/intuitions can only be found in cited references, for example, the conceptual explanation of the generalized random walk operator in Sevi et al. (2022); the diffusion distance in Coifman & Lafon (2006).  It would be better to add further notions and intuition in this paper, either main text or appendix.

There are also a number of important definitions, such as $l^2(V, v)$, delta function, the norm in the diffusion distance, etc, that can only be found in the two cited, which makes reading uneasy for the reader. On the other hand, for proposition 3.1 and its proof, the conclusion itself is clear already and the following proof is not  necessary.

- There are not sufficient discussions in Section 4.3, where the authors first formally introduce the parametrized diffusion distance and random walk diffusion kernel, which are the key components in their algorithm. It is not straightforward to see why one can extend the RWDK to the parameterized one without any restriction. For example, RWDK clustering and the spectral cluster have similar characteristics. Does this still hold for the parameterized version, i.e., P-RWDK?

- The algorithm requires a vertex measure as input for constructing the generalized random walk operator. There is not sufficient discussion about whether/how the choice of vertex measure influences the clustering performance, theoretically or empirically. For example, how bad the algorithm could perform on an arbitrarily chosen vertex measure?

- It would have been good to compare/contrast with some other very recent lines of work from this digraph literature

    x Higher-order spectral clustering of directed graphs, Steinar Laenen, He Sun (NeurIPS 2020)
    x Hermitian matrices for clustering directed graphs: insights and applications, Mihai Cucuringu, Huan Li, He Sun, Luca Zanetti (AISTATS 2020)

**Summary Of The Paper:**

This paper proposes a novel directed graph clustering method. which is mainly based on the so-called “diffusion geometry” from the diffusion maps/manifold learning/nonlinear dim reduction literature, and a generalized spectral clustering approach. To be more specific, the authors revisited the diffusion distance under the undirected setting by formulating it as Mahalanobis distance, based on which they constructed the random walk diffusion kernel (RWDK) and then showed the conceptual connection between RWDK clustering and the eigendecomposition of the classical spectral clustering method. Next, the authors proposed their directed graph clustering algorithm, namely parametrized random walk diffusion kernel (P-RWDK) clustering. This new method is a generalization on the aforementioned RWDK for clustering directed graphs and is obtained by replacing the random walk Laplacian with a parametrized random walk operator (P-RW).

For applying the proposed new algorithm, the authors provide a design on the parametrized random walk operator as well as using Calinski–Harabasz criterion for determining the diffusion time.  The authors performed experiments on multiple datasets (digraphs obtained from high-dimensional data using graph construction procedures, and real-world graphs) and demonstrated the effectiveness of their approach by comparing it with several other popular clustering algorithms.

Main Contributions:
- Under the undirected graph setting, the authors provide a new interpretation on the diffusion distance, and show the equivalence of the diffusion distance and the proposed random walk diffusion kernel (power of the transition matrix normalized by the vertex degrees).
- They also introduced a generalized random walk diffusion kernel by replacing the random walk Laplacian with the parameterized random walk operator, which might inspire other studies on diffusion geometry of directed graphs.
- Based on the above analysis, the authors design a new algorithm for digraph clustering as well as principled methods for fine-tuning the optimal algorithm parameters.
- The authors perform experiments on both synthetic and real-world data to compare their algorithm performance with others.


**Summary Of The Review:**

Interesting method, with good theoretical support, strong performance on numerical experiments, though the set of baselines could be expanded.

---

### Decision · Program_Chairs · 2023-01-20

**Decision:**

Reject

**Justification For Why Not Higher Score:**

As I have stated in the meta-review and the AC-reviewer meeting report, the lack of rigorous empirical evaluation is crucial as such evaluation may change the conclusion of the paper. Therefore this paper is not ready for publication at the moment. Since the core idea of this paper is still interesting, I recommend revising the paper according to reviewers' comments and re-submitting it.


**Justification For Why Not Lower Score:**

N/A

**Metareview: Summary, Strengths And Weaknesses:**

This paper studies clustering of nodes of directed graphs via random walks, and proposes the parametrized random walk operator that can be directly applied to directed graphs without any symmetrization, with which one can obtain the kernel gram matrix of nodes. Moreover, this paper aligns this operator with the diffusion geometry and draws a connection between them. The resulting clustering algorithm, called P-RWDKC, is empirically evaluated on real-world datasets.

### Strength

- The proposed technique of applying random walk operator to directed graphs without symmetrization can be an effective and promising approach.

- The connection between the random walk operator and the diffusion geometry presented in this paper is interesting and could inspire further development of this field.

### Weakness

- The technical novelty of this paper is not so high as reviewers pointed out because the proposal is a simple modification of the existing technique. Of course, it should be totally acceptable if such a simple strategy leads to interesting theoretical results and/or empirical advances, this paper does not meet the bar in terms of both aspects in my opinion.

- Presentation is sometimes unclear and more improvement is required. Although most of the unclear descriptions pointed by the reviewers have been resolved in the revised version, the paper still requires (at least) one more round of revision. For example, in the revised version, there is a typo in Eq.(4) or Eq.(1) as they are exactly the same in this version.

- Empirical evaluation is not thorough. As Reviewer xrEQ pointed out, since the performance of the proposal may be largely affected by the vertex measure $\nu$ and the diffusion time $t^*$, and parameter tuning is fundamentally difficult in the unsupervised setting, the sensitivity with respect to changes of them should be carefully examined. I agree with Reviewer xrEQ that experiments on synthetic examples generated by a directed SBM should be performed to examine it.

- Although I acknowledge the additional experiments about comparison with HERM and SC-SYM in the authors' response, more careful analysis and discussion is required in the main paper as this is an important comparison, especially HERM. It is currently unclear why their scores are so low compared to the proposal.


**Summary Of Ac-Reviewer Meeting:**

Since this paper is on a borderline, I held the AC-reviewer meeting. All the reviewers joined, and we have extensively and carefully discussed pros and cons of this paper while considering all the authors' responses and the updated paper.
Although we agree that this paper is well-motivated, mostly well-written, and proposes an interesting idea, several concerns were raised from reviewers, which are summarized as weaknesses in the above meta-review. In particular, we have agreed that the lack of evaluation of parameter sensitivity on synthetic data is crucial. Also, lack of careful comparison with HERM significantly deteriorates the quality of this paper. Since there is no theoretical assessment of the performance such as statistical analysis in this paper, rigorous empirical evaluation is indispensable. We therefore have agreed with rejecting the paper.